# Giving Families a Voice for Equitable Healthy Food Access in the Wake of Online Grocery Shopping

**DOI:** 10.3390/nu14204377

**Published:** 2022-10-19

**Authors:** Gabriela M. Vedovato, Shahmir H. Ali, Caitlin M. Lowery, Angela C. B. Trude

**Affiliations:** 1Institute of Health and Society, Federal University of São Paulo, Santos 11015-021, SP, Brazil; 2Department of Social and Behavioral Sciences, School of Global Public Health, New York University, New York, NY 10003, USA; 3Department of Nutrition, University of North Carolina at Chapel Hill, Chapel Hill, NC 27599, USA; 4Department of Nutrition and Food Studies, New York University Steinhardt School of Culture, Education, and Human Development, New York, NY 10003, USA

**Keywords:** healthy food access, food insecurity, health disparities, food assistance, grocery stores

## Abstract

Understanding the views of families from low-income backgrounds about inequities in healthy food access and grocery purchase is critical to food access policies. This study explored perspectives of families eligible for the Supplemental Nutrition Assistance Program (SNAP) on healthy food access in physical and online grocery environments. The qualitative design used purposive sampling of 44 primary household food purchasers with children (aged ≤ 8), between November 2020–March 2021, through 11 online focus groups and 5 in-depth interviews. Grounded theory was used to identify community-level perceived inequities, including influences of COVID-19 pandemic, SNAP and online grocery services. The most salient perceived causes of inequitable food access were neighborhood resource deficiencies and public transportation limitations. Rural communities, people with disabilities, older adults, racially and ethnically diverse groups were perceived to be disproportionately impacted by food inequities, which were exacerbated by the pandemic. The ability to use SNAP benefits to buy foods online facilitated healthy food access. Delivery fees and lack of control over food selection were barriers. Barriers to healthy food access aggravated by SNAP included social stigma, inability to acquire cooked meals, and inadequate amount of monthly funds. Findings provide a foundation for policy redesign to promote equitable healthy food systems.

## 1. Introduction

Inequities in healthy food access and diet quality drive disparities in food insecurity and obesity rates in the U.S., and are persistent problems in both urban and rural areas [1]. Racially/ethnically diverse, low-income and socially marginalized populations are significantly more likely to be food insecure and obese than white, higher-income populations [2,3]. Inequities in healthy food access are explained by systemic structural issues, such as poor-quality living conditions, lack of neighborhood resources and limited healthy food availability, which together influence food acquisition, eating behaviors, and health outcomes [4,5,6].

The COVID-19 pandemic has exacerbated employment and income disparities, exposed the vulnerability of food systems, and intensified inequities in access to healthy foods [7,8]. In an effort to mitigate the socioeconomic impact of COVID-19 in the U.S., large-scale government efforts such as the American Rescue Plan Act (ARP) which provided $1.9 trillion in aid to low-income families, including a temporary expansions of the Child Tax Credit of up to $3600 per child < age 5 and $300 per child < 18, stimulus relief checks, and meal distribution sites [9]. In addition, revisions to existing food and nutrition assistance programs of the U.S. Department of Agriculture (USDA) were made, including the temporary increase in benefits of the Supplemental Nutrition Assistance Program (SNAP) [10], more flexibility to the Special Supplemental Nutrition Program for Women, Infants and Children (WIC) requirements [11], implementation of the Pandemic Electronic Benefit Transfer (P-EBT) to supplement school meals [12], and the Farmers to Families Food Box Program (Food Box Program) [13,14].

Moreover, the rapid expansion of the SNAP Online Purchasing Pilot (OPP) allowed for the use of SNAP benefits to purchase groceries online [15], overcoming barriers related to physical access to healthy food among families in underserved areas with limited access to reliable transportation [16]. However, several deterrents and challenges to accessing online grocery services have been reported in the literature [16,17]. Examples include: limited availability of online grocery delivery services in low-income/low-access and rural areas [18], limited access to and comfort with use of the Internet and related technology [19], high perceived costs, and the paucity of deals online [20,21], concerns about control over grocery selection, the potential for losing money on unsatisfactory purchases, and the quality of selected fresh produce and meats [19].

Understanding low-income families’ perspectives on causes, barriers, and facilitators of healthy food access and grocery purchase is critical to ensure that policies are implemented to address the needs of underserved communities, and ultimately to promote equity in healthy eating [16]. The inclusion of families’ voices in the development of food policy is a powerful participatory approach to community capacity building and empowerment [5]. Few studies examining issues related to inequities in food access incorporate the voice of communities for a deeper understanding of perceived issues and immediate living conditions of families from low-income backgrounds [19,22,23,24]. Previous studies in urban Black neighborhoods adopted qualitative approaches to inform local food environment interventions, such as a community-owned food store [22] and a fruit and vegetable access program [23]. Recent research has focused on informing policy on nutrition assistance benefits for online grocery shopping. For instance, Zimmer et al. integrated the perspectives of WIC participants to design and implement a pilot test of WIC online ordering, ensuring compliance with policy regulations and improving the beneficiary’s shopping experience [19,24]. Although there has been an increase in studies on the use of SNAP benefits as a payment method for online groceries, more evidence is needed to understand underlying causes of inequitable food access, in order to promote equity through targeted approaches in the online and physical food retail environments [16].

The present study used qualitative research methods to examine community-level perceived inequities in healthy food access and how programs and services designed to improve food access, such as SNAP and online grocery shopping, may hinder or facilitate equity. By exploring perspectives of families eligible for SNAP in relation to healthy food access in physical and online grocery environments, this study aimed to answer the following research questions: (1) What are the perceived causes of inequity in access to healthy foods and who is mostly negatively impacted by it? (2) In what ways did the COVID-19 pandemic widen inequities in food access? and (3) How is access to healthy foods facilitated or hindered by programs such as SNAP or online grocery services?

## 2. Materials and Methods

Primary grocery shoppers of low-income households (≤130% of the federal poverty level and/or enrolled in SNAP) with children aged ≤8 living in Maryland were invited to participate in focus group discussions or in-depth interviews. Data collection occurred between November 2020 and March 2021 as part of a larger mixed methods study, details of which have been described elsewhere [25,26]. Briefly, participants were recruited via Facebook, ResearchMatch, and a community-based clinic to complete a survey. Purposeful sampling was used to ensure representation of individuals with and without online grocery shopping experience and SNAP participation in the past 12 months. Interested main household food purchasers (*n* = 95) were invited to attend a 60-min Zoom-based focus group. The interview guide was informed by the theory of planned behavior and previous literature on inequitable access to healthy foods to gather experiences with grocery shopping and the SNAP program [27,28]. Specifically, the interview guide (Appendix A) covered topics on attitudes, barriers and perceptions towards buying groceries online and in-store, reflections on food access and health equity, and suggestions to improve online grocery shopping and SNAP. Participants were asked to reflect upon both their own experiences as well as those of others in their communities in responding to the questions. If fewer than three participants attended a scheduled focus group, in-depth interviews were conducted. Forty-four individuals attended either a focus group (*n* = 39) or an in-depth interview (*n* = 5). Focus groups and interviews were audio-recorded and transcribed verbatim. Study procedures were approved by the University of Maryland School of Medicine Institutional Review Board.

Prior to attending a focus group or in-depth interview, all participants answered a brief Qualtrics survey that assessed study eligibility, participant’s sex, race/ethnicity, SNAP participation, previous online grocery shopping experience, and zipcode that was later coded according to the 2010 Rural-Urban Commuting Area (RUCA). Participants also responded to the 6-item validated USDA screening tool [29] that assessed food insecurity within the past 12 months. Validated cut-offs were used to categorize participants into low/very low food security and food security. 

Analysis of qualitative data was informed by principles of grounded theory [28] to explore inequities in access to healthy foods in the grocery environment by identifying the perceived (1) causes and (2) groups of people disproportionately experiencing these inequities. Inequities intensified by the COVID-19 pandemic emerged and were discussed by participants. A preliminary codebook was developed using information from inequities identified in past literature [20] and notes taken during data collection. The codebook was applied independently by two researchers on a set of transcripts and iteratively refined. Upon reaching 80% inter-coder reliability after double-coding two transcripts, the remaining 14 transcripts were independently coded. All analysis was performed using MAXQDA software [30].

## 3. Results

Most participants were female (91%), identified as Non-Hispanic Black (48%), lived in an urban area (88%), with income at or below 130% the Federal Poverty Line (91%), reported low or very low household food security (63%), and half have used online grocery services.

The emergent themes on perceived causes of inequitable food access and most affected groups in the context of COVID-19 and online grocery environment are presented in Appendix A.

### 3.1. Perceived Causes of Inequity in Access to Healthy Foods and Affected Groups

Income was commonly discussed as a perceived cause of inequity in healthy food access, such as unequal income distribution and high prices of healthy foods (illustrated in the quote below):


*“The pricing [of healthy food] is wrong. It’s ridiculous that’s causing a lot of obesity, because it’s cheaper for families that go out and get a bucket of chicken or a couple burgers at a fast-food place where they are $1 or less than it is to get something fresh, you know.”*
[Female, White/Caucasian, SNAP-participant]

Other perceived causes of inequities that emerged were associated with poverty and resource distribution at the community level. Issues related to transportation, such as a disconnected public transportation system or lack of car ownership, were barriers to physical grocery store access, whereas the inaccessibility of technology was a barrier to online grocery access. 

Race, disabilities, age, and gender emerged as important demographic factors linked with food inequities. Participants perceived older adults and people with disabilities to be disproportionately affected by the inaccessibility of foods (e.g., inability to carry heavy groceries, mobility related barriers). Participants also noted that while online grocery services may address transportation barriers, these groups may face additional challenges with digital literacy and ability to use technology. 

The relation between racial/ethnic groups and disparities in local food environments emerged in focus group discussions. The food environment of low-income neighborhoods, such as the disproportionate presence of convenience stores and unhealthy food options was perceived to be linked to racial and ethnic segregation of neighborhoods and social injustice. 

### 3.2. Ways That the COVID-19 Pandemic Has Widened Inequities in Food Access

The COVID-19 pandemic has had disparate impacts on the availability of healthy food in underserved communities, as reported in the following excerpts:


*“With COVID, my stores were out of so much that I didn’t honestly know how we were going to make it, because I had to buy things that were triple the price, just so we would have something to eat versus being hungry. And it wound up being a lot of microwave stuff and a lot of not actual food that didn’t last as long but was triple the price. And sometimes… every time somebody says “Oh, the COVID numbers are rising”, we get the same experience. So the stores are empty, the online stores are empty, and there’s (…) there’s not really any options, especially living in a small area.”*
[Female, White/Caucasian, non-SNAP participant]

Furthermore, the imminent risk of COVID-19 infection by crowded public transportation in urban areas was mentioned as a concern for in-store grocery shopping. Families with children from low-income backgrounds also reported facing more financial difficulties accessing healthy foods during COVID-19 pandemic than pre-pandemic. Some of these issues revolved around school closures and increased demand for quantity of foods at home to feed children during the pandemic. The closure of daycare centers and schools during the most acute stages of the pandemic also influenced loss of family income, as parents were no longer able to work or were forced to work from home and balance childcare responsibilities. 


*“So you have people who have lost their jobs, or because of COVID, and they’re not receiving help from the government, from the state. Everything’s at a standstill. What do they do? How do they feed themselves? Where do they get the money for food? They can’t work. They have small children who can’t go to daycare because it’s been closed. Can’t go to school because they’ve either completely closed down schools or they’re sending all the kids home to learn online.”*
[Female, White/Caucasian, non-SNAP participant]

### 3.3. Healthy Food Access Facilitated or Hindered by SNAP or Online Grocery Services

Perceptions of programs designed to address barriers to food access (i.e., SNAP—financial barriers and/or online grocery services—physical barriers) were interconnected with the role the policy or service played in addressing or exacerbating inequities (Appendix A). Although SNAP and online grocery shopping are two different means of improving access to food (i.e., government and private sectors, respectively), they intersect when the SNAP OPP was implemented. Participants acknowledged programmatic efforts to improve access to food, particularly with the expansion of the SNAP OPP and temporary additional benefits during the COVID-19 pandemic.


*“[SNAP] makes people be able to afford, afford not only…I mean afford healthy foods, I imagine that’s true. And on top of it, this whole being able to buy things online has got to make, for food deserts, it’s gotta make things better for people in food deserts. If people know they can, I’m assuming most people know now that they can get their food online.”*
[Female, White/Caucasian, SNAP-participant]

However, while participants noted that online grocery shopping and online SNAP benefits facilitated access to food for low-income communities, access to healthy foods specifically (such as fresh fruits and vegetables, or cooked meals) was still hindered by (i) cost (including delivery fees and tips from online grocery services not covered by SNAP) and (ii) mistrust of hired shoppers selecting quality items within online grocery shopping. For example, participants reported some healthy food items to be more expensive online.


*“For me, the bananas—I don’t mind the 99 cents (cost online), but that senior citizen might say, why am I gonna pay 99 cents when I can go to the store and get it for 59 cent in-person to save myself that 40 cents?*
[Male, Black/African American, SNAP-participant]

Participants reflected on potential negative unintended consequences of the SNAP program and of online grocery services in addressing food access. Specifically for SNAP, the perceived insufficient amount of monthly funds disbursed, paired with policies restricting use for ready-to-eat/cooked meals, and the small number of SNAP-authorized international markets hindered families’ ability to buy nutritious and culturally appropriate food in sufficient quantity and quality. 


*“Well me personally (…) they are going to give you food stamps but they don’t ever think how I am going to get to the store, where am I putting this food at if I don’t have a home, what am I doing with this food like I can’t get hot food (…) I am just, I gotta go to the market, every day, and just eat off the card every day, okay.”*
[Female, Black/African American, SNAP-participant]

In addition, difficulty in using the SNAP-EBT card and the social stigma that some SNAP participants experience when using the benefit when purchasing groceries in-store was highlighted:


*“Sometimes, it depends on the store, it depends on the area. Some corner stores in certain areas don’t accept [SNAP] EBT card because they couldn’t get approval, or someone scammed and caused a problem, complained. So they got shut down, so it is one less area for citizens of that community to access food. You need to come up with a way where you’re not discriminated in any way shape or form to where you can eat healthy foods for your family. You know, no, that is not asking for much, that is a basic need, food, you need food to survive.”*
[Female, White/Caucasian, SNAP-participant]

### 3.4. Causes and Consequences of Inequities in Healthy Food Access

Figure 1 summarizes a framework of emergent themes on causes and consequences of inequities in healthy food access (described further in Appendix A). The framework highlights the intersections between perceived causes of inequities and most negatively affected groups, including how SNAP and online grocery services may support or hinder equitable healthy food access. The confluence of structural and situational causes of food inequities (represented by the major rectangle), social resources (minor rectangles) and individual characteristics (circles) emphasize that some groups of people had different levels of vulnerability to limited healthy food access. The use of SNAP benefits and online grocery services were considered important factors that at times facilitated or—even if unintentionally—made it difficult for specific groups (based on race, age, gender, and disability) to access healthy foods.

## 4. Discussion

This qualitative research identified core inequities in healthy food access perceived by SNAP-eligible families that are critical to inform future policies and programs to support healthy physical and online grocery shopping. The conceptual framework that emerged from interviews reflected on how low-income families with children perceive causes of inequities, and the barriers to and facilitators of healthy food access for underserved populations. The emic framework [31] posits that the confluence of structural (e.g., food system inequities) and situational causes of inequities (e.g., COVID-19 pandemic) in healthy food access, linked to social and economic resources (e.g., income, technology access, transportation, locale) and demographic characteristics (e.g., race, age, disability, gender), can make some groups even more vulnerable to limited healthy food access.

The lack of neighborhood resources, higher prices of healthy foods compared to unhealthy foods, and lack of transportation and/or technology were perceived as key factors negatively impacting equity in healthy food access. The food environment can be experienced differently by groups with lower education, employment, socioeconomic status and residents in less disadvantaged neighborhoods, which will lead to different levels of exposure and vulnerability to unhealthy eating [32]. In this study, groups identified as the most negatively impacted by these inequities were rural communities, people with disabilities, older adults, and racially and ethnically diverse groups, corroborating studies that relate social markers of identity to inequities in healthy eating [2,3,6]. There is a need for special attention to racial and ethnic inequities in healthy food access [2,3], as communities of color are more vulnerable to unhealthy environments due to structural racism that has shaped policy and disinvestment, which must be tackled to disrupt a pattern that perpetuates in the U.S. [6].

Participants of this study echoed findings from other quantitative investigations that the COVID-19 pandemic has amplified existing disparities in access to healthy food, mainly due to job losses, local store closures and rising food prices [7,8]. The pandemic as a situational factor has been exposing previously existing social and economic weaknesses and exacerbated difficulties in accessing healthier foods at the community level, disproportionately affecting communities of color in the U.S. [7,8]. As a consequence, there is a need for food justice approaches and community empowerment initiatives to address disparities in access to healthy food [7]. For instance, O’Hara & Toussaint (2021) have described efforts in the face of COVID-19 pandemic to promote sustainable alternatives through community-centered strategies and cooperative business models that meet local food security needs [7].

The SNAP program (including the SNAP OPP) is designed to facilitate access to healthy foods [14,15]. However, the literature reveals unintended consequences for underserved groups [17,20,33,34,35,36,37], such as high fees and limited information about SNAP usage on retailer sites [17]. This qualitative study revealed that social stigma for families shopping at the store, inability to use benefits towards cooked meals, and inadequacy of monthly funds were barriers related to SNAP that hindered access to culturally and nutritionally appropriate foods. The recent reevaluation of the Thrifty Food Plan that resulted in an average increase in 25% to SNAP-pre-pandemic benefits is a step in the right direction, although more is needed. For instance, there is a need to reshape SNAP for rural participants, to reduce barriers to using SNAP EBT at rural farmers markets, and reformulate educational activities to address transportation-related barriers many rural families face [36]. Additionally, SNAP Plus Act of 2021 (H.R. 6338) has recently allowed states in the US to authorize SNAP benefits as payment methods for hot foods from approved restaurants and delis. This bill (named Restaurant Meals Program) currently applies to older adults, and individuals with disability or experiencing homeleness and is another positive development of the SNAP program to address the needs of underserved populations. However, findings from this study highlight the need to expand the restaurant Meals Program to other SNAP participants, like families with young children, who are at high risk for food insecurity.

Regarding the unintended consequences of online grocery services for underserved communities, our findings corroborate previous research on online grocery shopping behaviors among low-income shoppers [20,37]. Lower-income families seem less likely to shop for groceries online than higher-income households [37], and delivery fees have been a major barrier to online grocery use in many studies [16,20,21]. Likewise, in this study, additional fees related to delivery and tipping, and mistrust of food selection online were barriers to uptake of this service and the purchase of healthier, fresh foods. Disparities in technology access, digital literacy [38], and limited delivery services in low-income/low-access rural areas [18] are also current issues related to inequities in online grocery shopping.

Some limitations of the present study should be mentioned. The sample of low-income families eligible for SNAP with children (mainly women) recruited via convenience sampling. Other populations were not included; therefore, findings are not generalized to other low-income groups nationally. However, close to 50% of SNAP participants are households with children under the age of 18 years. Future studies should examine views of other groups, such as immigrants, older adults, or people with disabilities, who may have different perceptions of factors that hinder or facilitate equitable access to healthy foods. Possible selection bias towards those with access to the Internet and have familiarity with the technology to join a virtual focus group discussion, speak English, as well as those with available time, may exist. 

Participants’ views on healthy food inequities in underserved communities help to identify challenges and opportunities to alleviate structural inequities and address food insecurity and obesity among vulnerable populations. Understanding the perspective of primary shoppers of low-income SNAP eligible families allows a closer approximation of the lived experience of specific groups in relation to perceived barriers in accessing healthy foods both in physical and online grocery environments.

Public health educational programming should focus on participatory approaches that incorporate the voices of vulnerable groups in the process of social transformation to promote food as a human right [5]. Enabling the community to reflect on inequities and engage more directly to improve access to healthy foods is critical to promoting equitable food systems and food justice.

It is worth recognizing that the updated SNAP benefits linked to the SNAP OPP indicate advances towards more equitable food access for low-income communities [14,15]. However, some unintended consequences still need to be addressed to overcome barriers for more vulnerable groups. For example, SNAP OPP could be more sensitive to specific issues and demands to address local food needs to promote culturally and nutritionally appropriate food practices.

During the pandemic, online grocery services acted as a safety-net for low-income families to acquire food as an alternative or supplemental resource to physical grocery stores [8]. Furthermore, although SNAP was generally viewed as a positive program, many families stressed dissatisfaction with the inability to buy ready-to-eat/cooked meals with government benefits and the social stigma experienced with using SNAP in the physical stores. The SNAP OPP has the potential to decrease social stigma experienced by participants, as payment methods are unknown to hired online shoppers, although this feature was not salient to participants yet. 

## 5. Conclusions

In conclusion, our findings support further efforts to understand diverse perspectives of the root causes and social markers of identity linked to inequitable access to healthy food. Although online grocery services when intersected with the government benefit (SNAP) during the SNAP OPP were positively perceived by low-income families as a means to improve physical access to foods, many barriers remain that hinder financial access to healthy foods. The study provided a conceptual framework that can be used in policy/program design, advocacy and further research on approaches and strategies to promote equity in access to healthy foods. 

## Figures and Tables

**Figure 1 nutrients-14-04377-f001:**
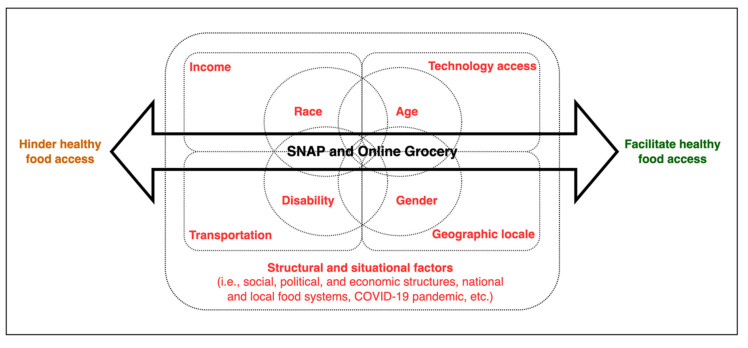
Conceptual framework depicting relationships between the perceived causes of inequities in healthy food access (rectangles), the most affected groups (circles), and the double-edged effect of SNAP and online grocery services supporting or hindering equity (two-sided arrow).

## Data Availability

The data that support the findings of this study are available on request from the corresponding author.

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
