# Peer review of "Giving Families a Voice for Equitable Healthy Food Access in the Wake of Online Grocery Shopping"

_nutrients, 2022, doi:10.3390/nu14204377_

Round 1

Reviewer 1 Report

This paper describes a qualitative study that was done with a large sample of participants to explore the perceived causes of inequalities in food access, how COVID has widened these inequalities and how SNAP and online grocery shopping can facilitate food access. Interesting paper, the Figure on the causes of inequalities in access is very interesting.

Specific comments

Methods section: More detail on the themes covered in the focus groups should be included, The interview guide could be provided as supplementary mateiral also.

Results section: It would have been interesting to have more detail on the sample characteristics. Figure 1 is very interesting but comes early and needsd ot be better linked with the emergent themes, At the moment, is it not clear how the framework emerged from the themes.

Results section: Why is SNAP and online grocery services treated in the same section? These are tow ways to improve access but there are many other ways....why were these two specifically chosen and why analys them together when they are so different in the sources of funding aim and objectives?

Discussion section: Last paragraph, the data presented does not necessarily support that online grocery services are a promising strategy to improve access...why not note this in the discussion?

Author Response

Response to Reviewer 1 Comments (Nutrients-1903734)

Point 1: Methods section: More detail on the themes covered in the focus groups should be included, The interview guide could be provided as supplementary mateiral also.

Response 1: We have added additional information on the specific types of questions covered in the interview guide, and have also attached the interview guide as a supplementary file (Box S1). Specifically, the interview guide (Box S1) covered topics on attitudes and perceptions towards buying grocery online, barriers of buying online and perceptions on attitudes towards health equity, and suggestions to improve online grocery shopping.” (lines 109-112)

Point 2: Results section: It would have been interesting to have more detail on the sample characteristics. 

Response 2: We added more information about the sample characteristics: Most participants were female (91%), identified as Non-Hispanic Black (48%) , lived in an urban area (88%), with income at or below 130% the Federal Poverty Line (91%),reported low or very low household food security (63%), and half have used online grocery services. (lines 139-141)

Point 3 Figure 1 is very interesting but comes early and needsd ot be better linked with the emergent themes, At the moment, is it not clear how the framework emerged from the themes.

Response 3: Thank you for the suggestion. The figure was moved to the end of the results section, and we improved the connection with emerging themes. Results - topic 3.4. Causes and consequences of inequities in healthy food access

Figure 1 summarizes a framework of emergent themes on causes and consequences of inequities in healthy food access (described further in Table S2 and Table S3). The framework highlights the intersections between perceived causes of inequities and most negatively affected groups, including how SNAP and online grocery services may support or hinder equitable healthy food access. The confluence of structural and situational causes of food inequities (represented by the major rectangle), social resources (minor rectangles) and individual characteristics (circles) emphasize that some groups of people had different levels of vulnerability to limited healthy food access. The use of SNAP benefits and online grocery services were considered important factors that at times facilitated or - even if unintentionally - made it difficult for specific groups (based on race, age, gender, and disability) to access healthy foods. (lines 278-300)

Point 4: Results section: Why is SNAP and online grocery services treated in the same section? These are tow ways to improve access but there are many other ways....why were these two specifically chosen and why analys them together when they are so different in the sources of funding aim and objectives?

Response 4: Thank you for raising this important point. We agree that SNAP and online shopping are two different means to improve access to food. However, our analysis highlights the intersection between them when the SNAP OPP was implemented. The two factors (online grocery service and SNAP) were chosen to demonstrate how each factor alone or in combination (SNAP OPP) acts as a barrier/facilitator to food access. We added a sentence in the results (topic 3.3.): Although SNAP and online grocery shopping are two different means of improving access to food (i.e., government and private sectors, respectively), they intersected when the SNAP OPP was implemented. (lines 242-244)

Point 5: Discussion section: Last paragraph, the data presented does not necessarily support that online grocery services are a promising strategy to improve access...why not note this in the discussion?

Response 5: Thank you for the comment. We made some adjustments to the paragraph to reflect the barriers associated with online grocery shopping that emerged from this investigation: “During the pandemic, online grocery services acted as a safety-net for low-income families to acquire food as an alternative or supplemental resource to physical grocery stores [8]. Furthermore, although SNAP was generally viewed as a positive program, many families stressed dissatisfaction with the inability to buy ready-to-eat/cooked meals with government benefits and the social stigma experienced with using SNAP in the physical stores. The SNAP OPP has the potential to decrease social stigma during in-person grocery shopping, as payment methods are unknown to hired online shoppers, although this feature was not salient to participants yet. Despite the benefits of online grocery shopping to address physical barriers imposed by the lack of access to physical grocery stores, participants voiced the need to promote greater control over food selection, strategies for shopping online on a budget, more payment options (e.g., cash at delivery, or use of multiple payment methods), and cost-saving options (e.g. reduced delivery and services fees for low-income customers) could improve accessibility of healthy foods online.” (lines 397-410)

Reviewer 2 Report

Thank you for the opportunity to review this manuscript, examining the perspectives of SNAP participants regarding equity in healthy food access. This topic is of high importance and the study addresses an important literature gap, with there currently being little available evidence as to the perceptions of online grocery retail users of lower socioeconomic status

I have outlined a few comments to help clarify several aspects of the manuscript:

Abstract/overall:

·        Very interesting study overall. Research on the online food retail environment is lacking in academic literature, so it is good to see a robust study.

·        A change of title may better reflect the scope of the study; as it stands it reads as though the study is focused on online grocery shopping but the paper is much broader than that and online grocery is a secondary focus

·        The focus on online grocery environment is inconsistently applied throughout the paper

Introduction:

·        Lines 43-44, a brief explanation of what the American Rescue Plan Act (ARP) and the Child Tax Credit are would be helpful for international readership

·        Line 57, suggest adding a brief definition of a food desert as this term is mentioned throughout.

Results:

·        Lines 148-153, this quote seems to be speaking more to food prices than it is to income. I would rephrase the mention of income in relation to this quote.

·        Lines 168-170, this point about those who live rurally or in food deserts would be more appropriate in the paragraph about “perceived causes of inequity” rather than in the COVID-19 paragraph

·        Lines 179-187, was this not the case before the COVID-19 pandemic though? This quote doesn’t really illustrate how COVID specifically has changed the availability of full service supermarkets in African American or Latino neighbourhoods

·        Lines 194-199, this quote could also be discussing how having daycares and schools closed may also mean loss of income to the family since parents are no longer able to work

·        Line 200, paragraph – There is no discussion here about how online grocery may facilitate or hinder healthy food access. Greater elaboration on this point is required as one of your research question is “How is access to healthy foods facilitated or hindered by programs such as SNAP or online grocery services?”

·        Lines 221-222, was there any discussion about how using SNAP online may reduce perceived stigma? Since cards do not need to be processed in person by an employee it may reduce the stigma experienced

Discussion:

·        Lines 264-267, was there discussion in your interviews about using SNAP online?

·        Line 280-282, these barriers to using online grocery were not discussed in your results. Would suggest adding participant comments surrounding these barriers to your results section.

Conclusion:

·        Line 324, there should be some mention of online grocery here to address your third research question

Author Response

Response to Reviewer 2 Comments (Nutrients-1903734)

Point 1: Abstract/overall: 

- Very interesting study overall. Research on the online food retail environment is lacking in academic literature, so it is good to see a robust study. 

-A change of title may better reflect the scope of the study; as it stands it reads as though the study is focused on online grocery shopping but the paper is much broader than that and online grocery is a secondary focus.

Response 1: Thanks to the reviewer for the comments. We changed the title in order to put online grocery shopping on the back burner, but not denying its relevance in the study. Tittle: Giving Families a Voice for Equitable Healthy Food Access in the Wake of Online Grocery Shopping (lines 2-3)

Point 2: - The focus on online grocery environment is inconsistently applied throughout the paper

Response 2: We revised the paper and made some changes in the discussion to improve the connection with the online grocery. environment. Discussion: changes to the first paragraph (“This qualitative research identified core inequities in healthy food access perceived by SNAP-eligible families that are critical to inform future policies and programs to support healthy physical and online grocery shopping.” - lines 302-304) 

Last paragraph (“During the pandemic, online grocery services acted as a safety-net for low-income families to acquire food as an alternative or supplemental resource to physical grocery stores [8]. Furthermore, although SNAP was generally viewed as a positive program, many families stressed dissatisfaction with the inability to buy ready-to-eat/cooked meals with government benefits and the social stigma experienced with using SNAP in the physical stores. The SNAP OPP has the potential to decrease social stigma experienced by participants, as payment methods are unknown to hired online shoppers , although this feature was not salient to participants yet. Despite the benefits of online grocery shopping to address physical barriers imposed by the lack of access to physical grocery stores, participants voiced the need to promote greater control over food selection, strategies for shopping online on a budget, more payment options (e.g., cash at delivery, or use of multiple payment methods), and cost-saving options (e.g. reduced delivery and services fees for low-income customers) could improve accessibility of healthy foods online. (lines 397-410)

Point 3: Introduction: -Lines 43-44, a brief explanation of what the American Rescue Plan Act (ARP) and the Child Tax Credit are would be helpful for international readership

Response 3: We added the following sentence: “In an effort to mitigate the socioeconomic impact of COVID-19 in the U.S., large-scale government efforts such as the American Rescue Plan Act (ARP) which provided $1.9 trillion in aid to low-income families, including a temporary expansions of the Child Tax Credit of up to $3,600 per child < age 5 and $3,00 per child < 18, stimulus relief checks, and meal distribution sites have been launched [9].” (lines 44-48)

Point 4: -Line 57, suggest adding a brief definition of a food desert as this term is mentioned throughout.

Response 4: The term food desert was replaced by "low-income and low-access" throughout to designate areas with limited access to healthy food to reflect what is measured in USDA Food Access Research Atlas (FARA). Introduction (lines 61-66): "Examples include: limited availability of online grocery delivery services in low-income/low-access and rural areas [18], limited access to and comfort with use of the Internet and related technology [19], high perceived costs, and the paucity of deals online [20,21], concerns about control over grocery selection, the potential for losing money on unsatisfactory purchases, and the quality of selected fresh produce and meats [19]."

Point 5: Results:- Lines 148-153, this quote seems to be speaking more to food prices than it is to income. I  would rephrase the mention of income in relation to this quote.

Response 5: We thank the reviewer for pointing this out. We rephrased the sentence in the results to highlight the issue of high food prices: “[...] high prices of healthy foods (illustrated in the quote below):” (line 146) 

Point 6: -Lines 168-170, this point about those who live rurally or in food deserts would be more appropriate in the paragraph about “perceived causes of inequity” rather than in the COVID-19 paragraph

Response 6: We thank the reviewer for this relevant observation. The paragraph and quote were revised and deleted because they were unrelated to COVID-19.

Point 7: -Lines 179-187, was this not the case before the COVID-19 pandemic though? This quote doesn’t really illustrate how COVID specifically has changed the availability of full service supermarkets in African American or Latino neighbourhoods

Response 7: We replaced it with a more illustrative quote on the impact of COVID-19 on the availability of food stores in underserved communities: “With COVID, my stores were out of so much that I didn't honestly know how we were going to make it, because I had to buy things that were triple the price, just so we would have something to eat versus being hungry. And it wound up being a lot of microwave stuff and a lot of not actual food that didn't last as long but was triple the price. And sometimes… every time somebody says “Oh, the COVID numbers are rising,” we get the same experience. So the stores are empty, the online stores are empty, and there's (..) there's not really any options, especially living in a small area.” [Female, White/Caucasian, non-SNAP participant] (lines 171-178)

Point 8: -Lines 194-199, this quote could also be discussing how having daycares and schools closed may also mean loss of income to the family since parents are no longer able to work

Response 8: Thank you for the suggestion, we added a sentence to discuss this point: The closure of daycare centers and schools during the most acute stages of the pandemic also influenced loss of family income, as parents were no longer able to work or were forced to work from home and balance childcare responsibilities. (lines 230-232)

Point 9: Line 200, paragraph – There is no discussion here about how online grocery may facilitate or hinder healthy food access. Greater elaboration on this point is required as one of your research question is “How is access to healthy foods facilitated or hindered by programs such as SNAP or online grocery services?”

Response 9: We have now added additional discussion and a quote to elaborate on how online grocery facilitated / hindered access to healthy food specifically: However, while participants noted that online grocery shopping and online SNAP subsidies on the whole facilitated improved access to food for low-income communities, access to healthy foods specifically (such as fresh fruits and vegetables, or cooked meals) was still hindered by cost and quality related barriers within online grocery shopping. For example, participants reported some healthy food items to be more expensive online. “For me, the bananas - I don't mind the 99 cents (cost online), but that senior citizen might say, why am I gonna pay 99 cents when I can go to the store  and get it for 59 cent in-person to save myself that 40 cents? [Male, Black/African American, SNAP-participant]” (lines 252-259)

Point 10: Lines 221-222, was there any discussion about how using SNAP online may reduce perceived stigma? Since cards do not need to be processed in person by an employee it may reduce the stigma experienced

Response 10: Social stigma in online grocery shopping was not the theme/issue of the study, but it was identified in our analysis. Thus, we improved the discussion of this topic: “Furthermore, although SNAP was generally viewed as a positive program, many families stressed dissatisfaction with the inability to buy ready-to-eat/cooked meals with government benefits and the social stigma experienced with using SNAP in the physical stores. The SNAP OPP has the potential to decrease social stigma during in-person grocery shopping, as payment methods are unknown to hired online shoppers, although this feature was not salient to participants yet.” (lines 399-404)

Point 11: Discussion: -Lines 264-267, was there discussion in your interviews about using SNAP online?

Response 11: Yes, there were discussions in the interviews/focus groups about using SNAP online. The main emergent themes were presented in Table S3 (Supplementary Materials) and in results (Topic 3.3.): "Although SNAP and online grocery shopping are two different means of improving access to food (i.e., government and private sectors, respectively), they intersect when the SNAP OPP was implemented. Participants acknowledged programmatic efforts to improve access to food, particularly with the expansion of the SNAP OPP and temporary additional benefits during the COVID-19 pandemic. “[SNAP] makes people be able to afford, afford not only…I mean afford healthy foods, I imagine that's true. And on top of it, this whole being able to buy things online has got to make, for food deserts, it's gotta make things better for people in food deserts. If people know they can, I'm assuming most people know now that they can get their food online.” [Female, White/Caucasian, SNAP-participant] (lines 242-250)

Point 12: -Line 280-282, these barriers to using online grocery were not discussed in your results. Would suggest adding participant comments surrounding these barriers to your results section.

Response 12: We added an additional paragraph and quote to address this point in the results: However, while participants noted that online grocery shopping and online SNAP subsidies on the whole facilitated improved access to food for low-income communities, access to healthy foods specifically (such as fresh fruits and vegetables, or cooked meals) was still hindered by cost (including delivery fees and tips from online grocery services not covered by SNAP) and mistrust of hired shoppers selecting quality items. For example, participants reported some healthy food items to be more expensive online. “For me, the bananas - I don't mind the 99 cents (cost online), but that senior citizen might say, why am I gonna pay 99 cents when I can go to the store  and get it for 59 cent in-person to save myself that 40 cents? [Male, Black/African American, SNAP-participant] (lines 251-259)

Point 13: Conclusion: -Line 324, there should be some mention of online grocery here to address your third research question

Response 13: We expanded the conclusions section to include mention of online grocery: In conclusion, our findings support further efforts to understand diverse perspectives of the root causes and social markers of identity linked to inequitable access to healthy food. Although online grocery services when intersected with the government benefit (SNAP) during the SNAP OPP were positively perceived by low-income families as a means to improve physical access to foods, many barriers remain that hinder financial access to healthy foods. The study provided a conceptual framework that can be used in policy/program design, advocacy and further research on approaches and strategies to promote equity in access to healthy foods. (lines 412-419)